# The clinical characteristics and SARS-CoV-2 infection in children of acute hepatitis with unknown aetiology: A meta-analysis and systematic review

Jiayi Shan[1]☯, Baoyi Huang[2]☯, Yijun Xin[2], Ran Li[2], Xiaoling Zhang[2], Hua Xu[3]*

1 Department of Pediatrics of Traditional Chinese Medicine, Guangzhou Women and Children's Medical Center, Guangzhou, China, 2 The First Clinical Medical School, Guangzhou University of Chinese Medicine, Guangzhou, China, 3 Department of Pediatrics, The First Affiliated Hospital of Guangzhou University of Chinese Medicine, Guangzhou, China

☯ These authors contributed equally to this work.
* xuhua0212@163.com

**Data Availability Statement:** All raw data are available from the public repository database

## Abstract

The World Health Organization has issued a global alert on Acute Severe Hepatitis of Unknown Aetiology (AS-HEP-UA) since 23 April 2022,and there was still uncertainty regarding the association of AS-HEP-UA with SARS-CoV-2 as well as adenovirus. This study aimed to summarize the infection of SARS-CoV-2 and co-infections with adenovirus, as well as clinical features and outcomes in patients with AS-HEP-UA. PubMed, Embase, Web of Science, and the Cochrane Library were searched from 1 October 2021 to 8 December 2022 for studies about patients with AS-HEP-UA. This study was registered in the PROSPERO database (CRD42023385056). We has included 14 eligible articles. The main clinical features of AS-HEP-UA were jaundice (65%) and vomiting (59%), while other clinical features included diarrhea (45%), abdominal pain (37%), and fever (31%), roughly 10% of the children required liver transplantation. The overall positivity rate for SARS-CoV-2 was 21.6% (95% CI: 0.126–0.319), with 25.5% (95% CI: 0.161–0.358) for previous infections. The positivity rate for adenovirus infection was 58.6% (95% CI:0.429–0.736) while co-infection with SARS-CoV-2 was 17.5% (95% CI: 0.049–0.342). Moreover, we found that the positive rate of SARS-CoV-2 for this hepatitis outbreak was correlated with region by subgroup analysis. In conclusion, the positive rate of adenovirus was higher than SARS-CoV-2, and the relationship between AS-Hep-UA and COVID-19 is not significant. However, it cannot be excluded that the COVID-19 epidemic is an indirect causative agent of AS-Hep-UA, which requires a larger cohort of AS-Hep-UA patients to uncover additional findings.

## Introduction

Since the report of 10 cases of severe acute hepatitis of unknown etiology in children was shown in central Scotland on 5 April 2022 [1], additional cases have been reported from the

(https://figshare.com/articles/dataset/minimal_data_set_xlsx/26643910).

**Funding:** The authors received no specific funding for this work.

**Competing interests:** The authors have declared that no competing interests exist.

European, the United States and other regions. Sine then, the World Health Organization (WHO)has issued a global alert on 23 April 2022 [2], causing widespread concern. As of September 25, 2022, 1063 probable cases have been notified worldwide [3–5]. Due to the rapid progression and unknown cause of the disease, the case is referred to as "Acute Severe Hepatitis of Unknown Aetiology" (AS-HEP-UA).

Initially, the case definitions of AS-Hep-UA varied between countries and time periods [6], but to collect data for analysis, by the end of April 2022, WHO and ECDC had started joint surveillance using the European surveillance system tesy(TESSy) [6]. A confirmed case is defined as a person presenting with an acute hepatitis (non-hepatitis viruses A, B, C, D and E) with aspartate transaminase (AST) or alanine transaminase (ALT) over 500 IU/L, who is 16 years old or younger, since 1 October 2021,excluding cases with a clear etiology.

The clinical manifestations of AS-HEP-UA are acute onset, including fatigue and loss of appetite, nausea, vomiting, diarrhea and other gastrointestinal symptoms [7]. However, for AS-HEP-UA patients, no direct cause was found [8]. It was found that the AS-HEP-UA outbreak developed during the ongoing COVID-19 pandemic commencing in 2019, with cases not arising until long after the peak of the epidemic, remaining a question that whether SARS-CoV-2 infection was the triggers of the AS-HEP-UA outbreak [6,9]. Beside adenovirus was also considered as a potential cause recently, but it did not fully explain the intensity of clinical picture [10].

Despite the clinical characters and lab indicators of AS-HEP-UA in individual regions has been reported previously reports, there is still uncertainty regarding the association of AS-HEP-UA with SARS-CoV-2 as well as adenovirus, lacking a comprehensive systematic review for pool analysis. Therefore, based on various reports of AS-Hep-UA cases worldwide, we decide to design a study to summarize the prevalence of recent or previous infection of SARS-CoV-2 and co-infections with adenovirus, as well as clinical features and outcomes in patients with AS-HEP-UA, which may have positive clinical and epidemiological implications for treatment and control, getting a more in-depth look at this hepatitis.

## Materials and methods

### Search strategy and study selection

Four databases including PubMed, Embase, Web of Science, and the Cochrane Library were systematically searched from 1 October 2021 to 8 December 2022 in line with the Preferred Reporting Items for Systematic Reviews and Meta-Analyses process. Following terms was used for searching by search strategy without language limitation: COVID-19; coronavirus; SARS-CoV-2; pandemic; children; pediatric; hepatitis; liver. The detailed search strategy is detailed in S1. In addition, relevant articles and references were included as eligible articles after screening. All articles were managed with Endnote (version X9.2).

### Inclusion and exclusion criteria

The inclusion criteria were as follows: (1) Population need to meet the WHO definition of hepatitis of unknown etiology in children: A person presents with acute hepatitis (non-hepatitis viruses A, B, C, D, and E) with AST or ALT over 500 IU/L, and is 16 years old or younger, since 1 October 2021. (2) Including data on COVID-19 and adenovirus testing. (3) Including the outcome and the clinical characteristics of the patients. (4) Study design: case series, observational study, randomized controlled trials and cross-sectional. The exclusion criteria were as follows:(1) Review. (2) No reported sample size. (3) Insufficient data.

## Data extraction and quality assessment

Two of the investigators (Xiaoling Liu and Ran Li) independently extracted data, including first author, publication time, sample size, area, study type, demographic features, clinical characteristics, laboratory indices and disease outcomes for meta-analysis, and possible disagreements were resolved by consensus with all investigators. We assessed the methodological quality of the included studies except case control studies using the Case Series Study Quality Assessment Tool published by the National Institutes of Health [11], which are consisted of eight items. We scored each item as 0 or 1 according to the criteria and ranked as low ($\geq$7), medium (5–6), or high risk of bias ($\leq$4) depending on the overall score by two reviewers (Yijun Xin and Jiayi Shan). However, for case control studies, we used the Newcastle-Ottawa Quality Assessment Scale for additional clarification by following 3 aspects: selection (4 stars), comparability (2 stars) and outcome (3 stars).We would regard 7 out of 9 stars as low risk of bias, 4–6 stars as moderate risk, and less than 4 stars as high risk of bias, details in S1 Table.

To include as many studies as possible in our systematic review, we systematically contacted authors or co-authors when information was missing in the full-text paper.

## Outcome

We selected endpoints based on the available outcomes from published articles. Our primary outcomes included viral positivity in patients with hepatitis, including rates of previous /current SARS-CoV-2 infection, adenovirus infection, and co-infection with SARS-CoV-2 and adenovirus.

Secondary outcomes included a wide range of clinical manifestations of hepatitis of unknown, including jaundice, vomiting, diarrhea, abdominal pain, fever, upper respiratory symptoms.

## Statistical analysis

We assessed statistical heterogeneity using the $I^2$ statistic and Cochran's Q test. A fixed-effects model was used when heterogeneity was considered to exist with $p<0.05$ or $I^2>50\%$. Or else, a random effects model was applied which were used to aggregate effect sizes for estimating overall effect estimates and corresponding 95% confidence intervals (CIs). In our study, subgroup analyses were performed according to the different regions reported in the article. All statistical analyses were performed with STATA software (version 16.0, Stata Corp. LP).

## Study registration

This study is available in the PROSPERO database (University of York, UK; http://www.crd.york.ac.uk/PROSPERO/) registered as CRD42023385056.

# Results

## Literature inclusion and characteristics

We collected a total of 10,541 potentially relevant studies, of which 10,507 were retrieved from the database by search strategy and 34 were identified as relevant from the references. After removing duplicate literature, 8101 articles remained for further screening. After browsing through the titles and abstracts, 34 potentially includable articles were selected for full-text browsing. After full-text evaluation, 14 studies were included in this systematic review and meta-analysis (Fig 1). Of these studies, 2 studies [12,13] were case series,6 studies [14–19] were observational study,4 studies [20–23] were cross-sectional and 2 studies [24,25] were case control (Table 1).

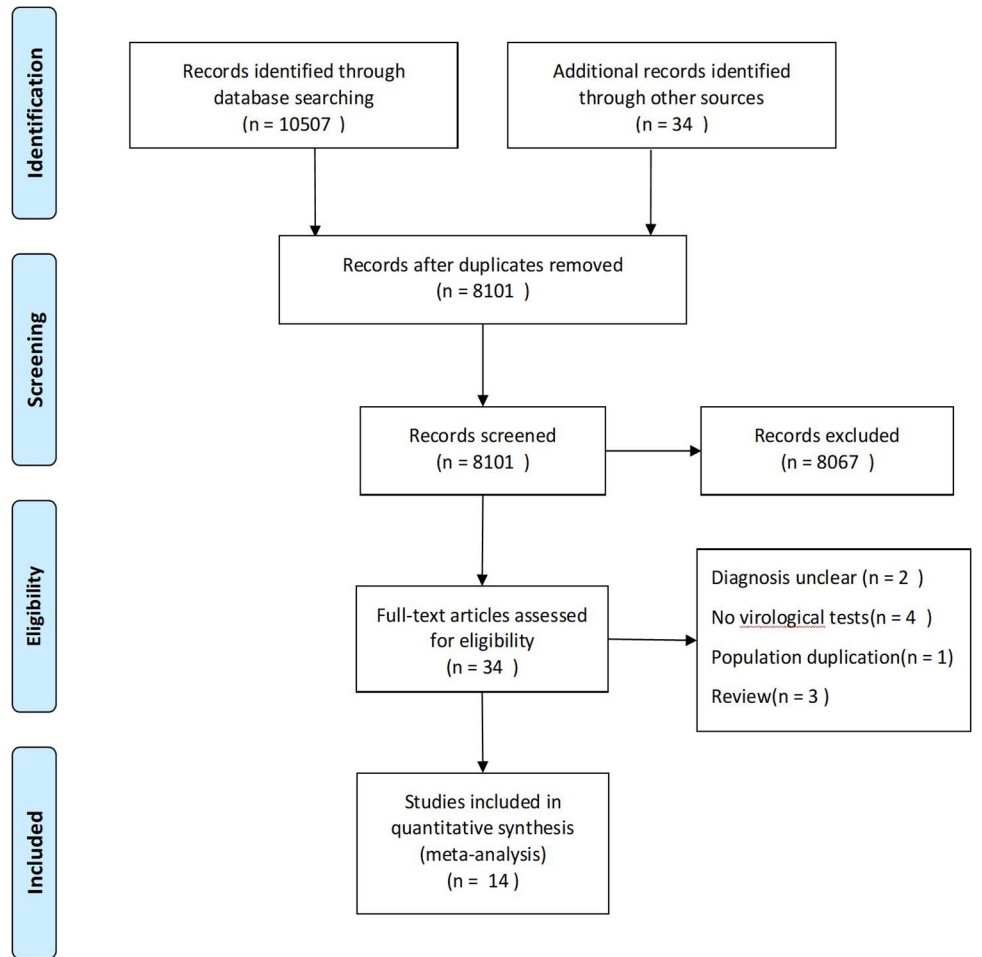

**Fig 1. Prisma flow diagram reporting the systematic review.**

## Clinical characteristics

The clinical characteristics were analyzed according to the included studies. As the results shown, the proportion of most common clinical characters were 0.656 (95% CI: 0.488-0.808) of jaundice and 0.591 (95% CI: 0.542-0.640) of vomiting in pediatric inpatients with acute hepatitis of unknown cause with heterogeneity of 86% and 0%, respectively. And the proportion of diarrhea was 0.454 (95% CI: 0.322-0.588), abdominal pain was 0.374 (95% CI: 0.304-0.446), fever was 0.314 (95% CI: 0.174-0.471), upper respiratory symptoms was 0.287 (95% CI: 0.119-0.490) (Fig 2) (Table 2).

## Associated virus infection

This article focuses on the analysis of infection status in pediatric inpatients with acute hepatitis of unknown cause. Among cases with at least one test result available for any pathogen, for SARS-CoV-2,0.216 (95% CI: 0.126-0.319) were positive while 0.255 (95% CI: 0.161–0.358) had previous infection of SARS-CoV-2. For adenovirus,0.586(95% CI: 0.429–0.736)were positive and 0.175 (95% CI: 0.049–0.342)were coinfected with both SARS-CoV-2 and adenovirus. Adenovirus and SARS-CoV-2 results were available from either whole blood, serum, respiratory, faecal or other/unknown sample type (Fig 3) (Table 3).

**Table 1. The characteristics of the included literatures.**

| First author | Study design | Study date | Area | Gender (%M) | Age median (IQR) | N | Quality score |
|---|---|---|---|---|---|---|---|
| Julia M.B. [12] | Case series | Oct 2021-Feb 2022 | Alabama USA | 78% | 2y11m (1y8m-5y9m) | 9 | 7 |
| Jordan C. [15] | Observational study | Oct 2021-Jun 2022 | USA | 41% | 2y2m (0y–9y7m) | 296 | 6 |
| Kelgeri C. [20] | Cross-sectional | Jan 2022-Apr 2022 | UK | 55% | 4y (1y–7y) | 44 | 7 |
| Willem S L. [14] | Observational study | Spring of 2022 | Netherlands | 80% | 2y (17m-3y) | 5 | 5 |
| Kimberly M. [16] | Observational study | Jan 2022-Apr 2022 | Scotland | 54% | 3y9m (3y6m-4y6m) | 13 | 7 |
| Pejman R. [13] | Case series | Mar 2022-May 2022 | Iran | 33% | 12y (6y-12y) | 3 | 6 |
| Adriana R V.(21) | Cross-sectional | Jan 2022-Jun 2022 | European | NM | NM | 427 | 5 |
| Elke W. [25] | Case control | Oct 2021-Jan 2022 | Leuven Belgium | NM | NM | 9 | 6 |
| UKHSA [22] | Cross-sectional | As of 4 July 2022 | UK | NM | NM | 274 | 5 |
| Antonia H. [24] | Case control | Mar 2022-Apr 2022 | Scotland | 56% | 3y9m (3y4m-5y1m) | 9 | 7 |
| Akash D. [18] | Observational study | Feb 2022-May 2022 | UK | NM | 2y10m (2y5m-4y5m) | 8 | 5 |
| Ruben H de K.(23) | Cross-sectiona | Jan 2022-Apr 2022 | European and Israel | 44% | 7y7m (28d–16y) | 26 | 4 |
| Anita V. [19] | Observational study | Jan 2022-May 2022 | UK | 44% | 3y2m (1y7m-8y9m) | 9 | 6 |
| Fabiola Di D.(17) | Observational study | Jan 2022-May 2022 | Italian | NM | 4y3m(1y-14y3m) | 34 | 6 |

Abbreviation: NM, not mentioned; N, number of population.

## Liver transplantation

Our analysis encompasses the outcomes of children who suffered from severe acute hepatitis of unknown etiology. Among them, 0.103 (95% CI: 0.047–0.173) underwent liver transplantation (Fig 4). In addition, according to the data we included in the article, there were 11 registered deaths and 47 with missing data.

## Publication bias

Regarding publication bias, the Eggers' test was conducted for all performances and mostly no statistically significant bias was found, except for SARS-CoV-2 infection (t = 2.22, P = 0.047) and liver transplantation (t = 2.24, P = 0.045) (S2 Table; Table 4).

## Subgroup analysis

Our main interest in this study was to investigate the relationship between AS-HEP-UA and SARS-CoV-2 infection, and the results of subgroup analysis for SARS-CoV-2 infection based on different regions are presented in our study (Fig 5). As depicted by the forest plots, a majority of cases were reported from the United Kingdom (43.3%) with an incidence rate of AS-HEP-UA as 0.381 (95% CI: 0.184–0.598). In contrast, non-UK regions exhibited lower incidence rates as 0.119 (95% CI: 0.030–0.243). Statistically, the incidence of SARS-CoV-2 infection is regionally related ($p = 0.031$) (Fig 5A). In addition, we performed a subgroup analysis of study type and found major heterogeneity in observational study ($I^2 = 81.816\%$, $p < 0.001$). Also, the incidence of SARS-CoV-2 infection was statistically found to be related to the type of study ($p = 0.023$) (Fig 5B).

## Discussion

Awareness of the clinical features and possible pathogens of AS-HEP-UA is not only important for clinicians, but also helps health policy makers to make more targeted preventive measures and holistic care models, which may help prevent the progression of novel hepatitis in

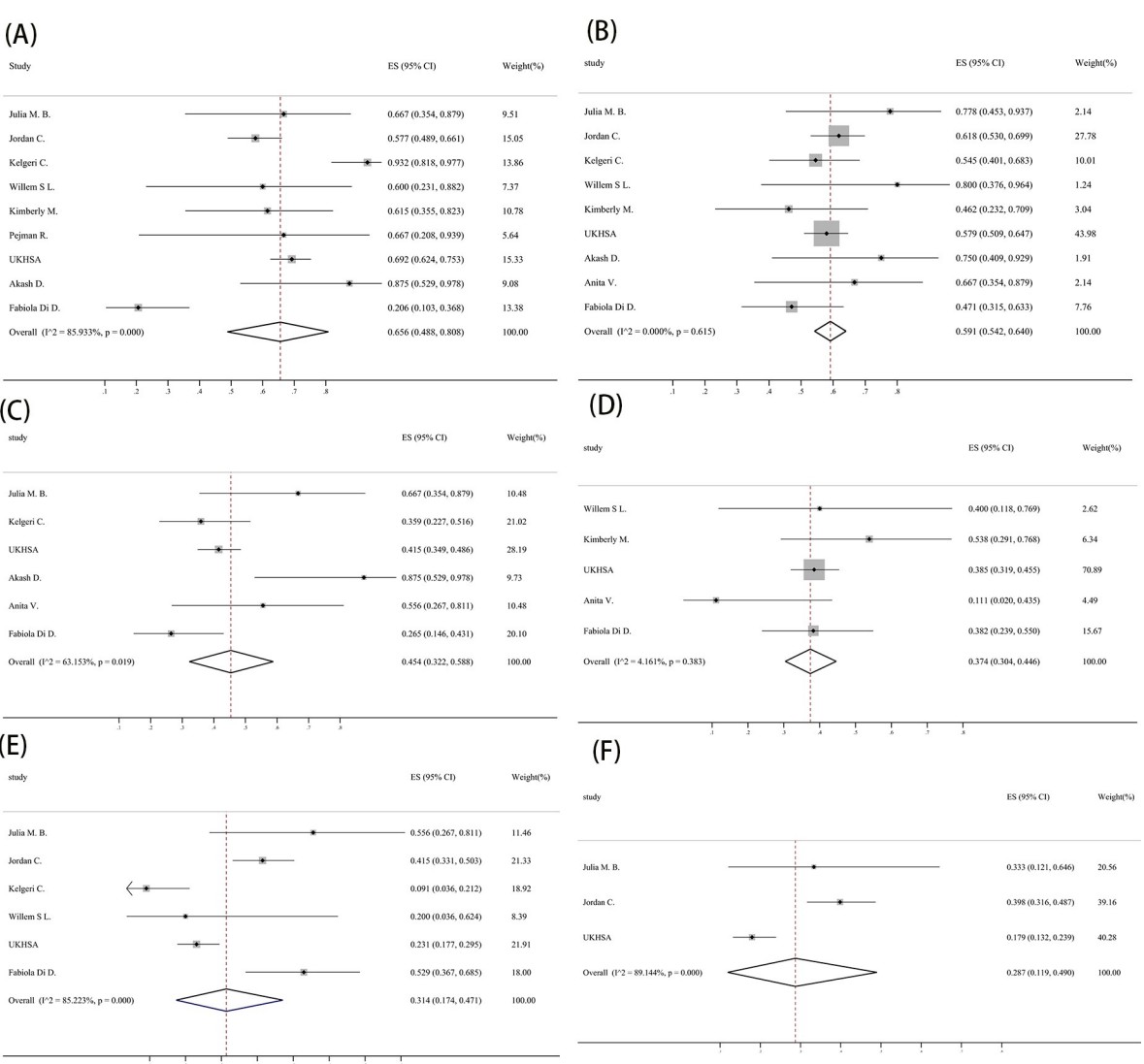

**Fig 2. Forest plots of clinical characters.** (A) jaundice;(B) vomiting;(C) diarrhea;(D) abdominal pain;(E) fever;(F) upper respiratory symptoms.

**Table 2. Meta-analysis results of clinical characters.**

| Variable | N | Estimate | 95% CI | I² (%) | P |
|---|---|---|---|---|---|
| jaundice | 434 | 0.656 | (0.488–0.808) | 86 | < .010 |
| vomiting | 440 | 0.591 | (0.542–0.640) | 0 | 0.615 |
| diarrhea | 294 | 0.454 | (0.322–0.588) | 63 | 0.019 |
| abdominal pain | 256 | 0.374 | (0.304–0.446) | 4 | 0.383 |
| fever | 410 | 0.314 | (0.174–0.471) | 85 | < .010 |
| upper respiratory symptoms | 327 | 0.287 | (0.119–0.490) | 89 | < .010 |

Abbreviation: CI, confidence interval.

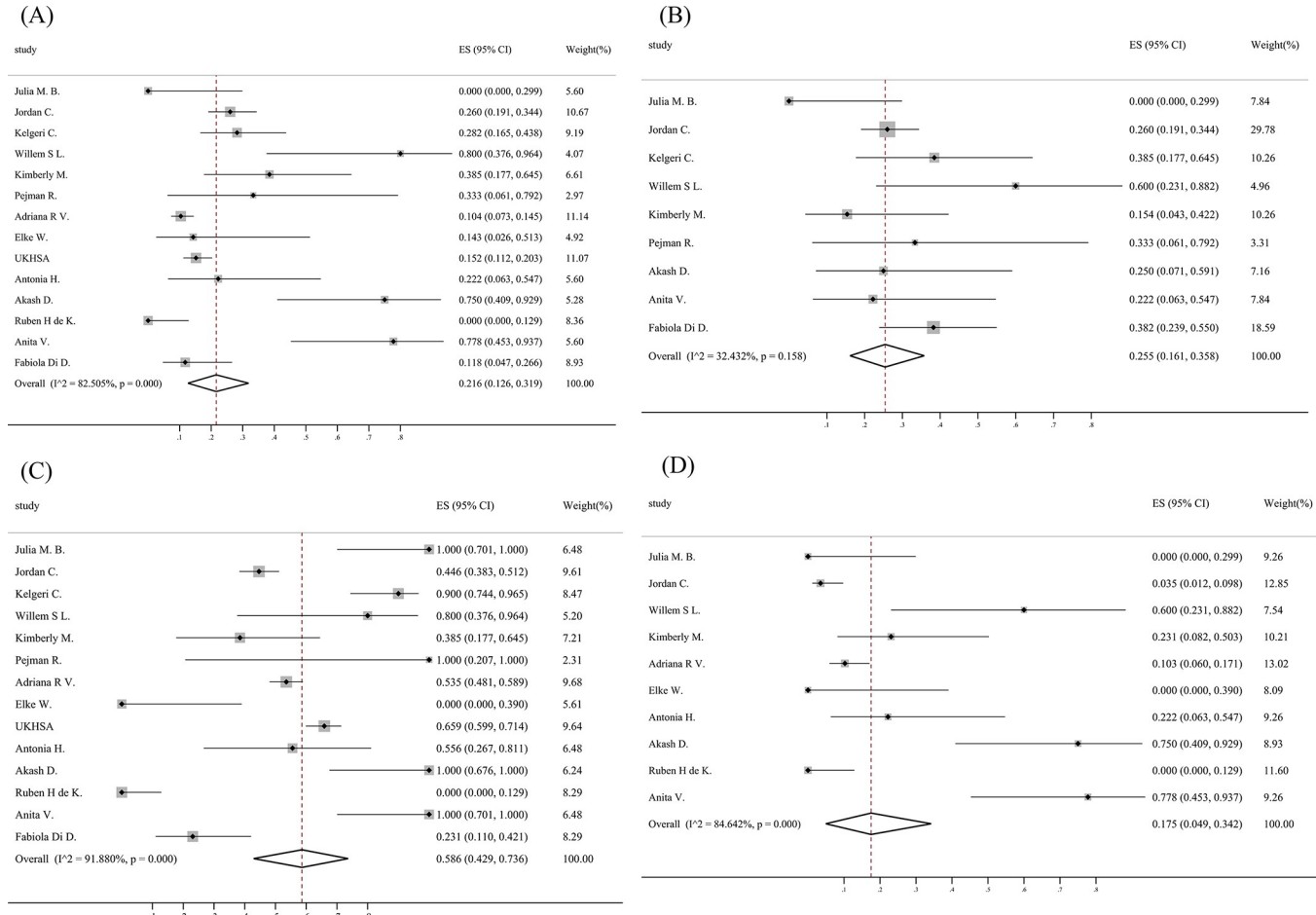

**Fig 3. Forest plots of associated virus infection.** (A) SARS-CoV-2; (B) previous infection of SARS-CoV-2; (C) Adenovirus; (D) SARS-CoV-2 & adenovirus.

patients. Therefore, this systematic review and meta-analysis was conducted to gain more insight into the AS-HEP-UA information.

In this systematic review and meta-analysis, we performed a pooled analysis for 14 eligible articles describing the epidemiological features as well as clinical symptoms and outcomes of AS-Hep-UA. For clinical symptoms, the main clinical features of AS-HEP-UA were jaundice (65%) and vomiting (59%), while other clinical features included diarrhea (45%), abdominal pain (37%), and fever (31%). From the analysis of the articles with precisely recorded outcomes, it was concluded that roughly 10% of the children required liver transplantation.

**Table 3. Meta-analysis results of associated virus infection.**

| Positive virus | N | Estimate | 95% CI | I² (%) | P |
|---|---|---|---|---|---|
| SARS-CoV-2 | 802 | 0.216 | (0.126–0.319) | 82 | < .010 |
| Adenovirus | 949 | 0.586 | (0.429–0.736) | 91 | < .010 |
| SARS-CoV-2&adenovirus | 288 | 0.175 | (0.049–0.342) | 84 | < .010 |
| previous infection of SARS-CoV-2 | 217 | 0.255 | (0.161–0.358) | 32 | 0.158 |

Abbreviation: CI, confidence interval; N, number of population.

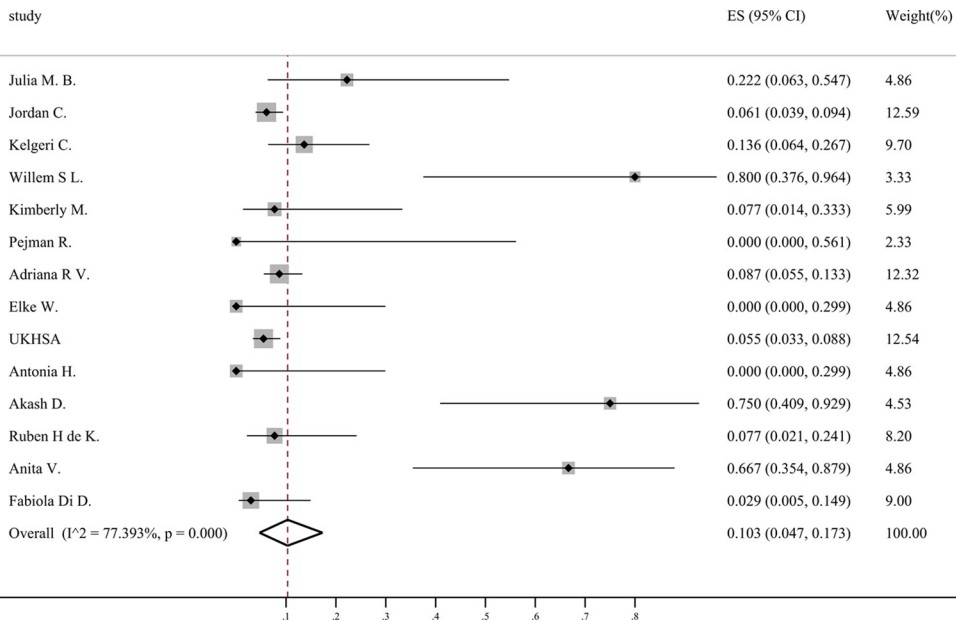

**Fig 4. Forest plots of liver transplantation.**

Among all included children, the overall positivity rate for SARS-CoV-2 was 21.6% (95% CI: 0.126–0.319), with 25.5% (95% CI: 0.161–0.358) for previous infections. The positivity rate for adenovirus infection was 58.6% (95% CI:0.429–0.736) while co-infection with SARS-CoV-2 was 17.5% (95% CI: 0.049–0.342). Moreover, we found that the positive rate of SARS-CoV-2 for this hepatitis outbreak was correlated with region by subgroup analysis, and the rate of SARS-CoV-2 infection was higher in the UK compared to non-UK regions. The reason for this regional difference is not clear at the moment, perhaps it is related to the proactivity of reporting in different regions, and so far we have not found any reports of AS-HEP-UA from Asia or Africa via our search.

An acute hepatitis study group found that the number of probable cases were elevated in 5/17 European and 1/7 non-European countries, with a significant increase in the number of severe cases in Europe among the 24 countries recorded during the period January 1 to April 18, 2022, versus the number of cases in the past 5 years [26]. Concordant with the our study,

**Table 4. Results of egger test.**

| Performance | t-value | p-value | 95% confidence interval |
|---|---|---|---|
| SARS-CoV-2 | 2.22 | 0.047 | (0.01–1.67) |
| SARS-CoV-2&adenovirus | 2.01 | 0.079 | (-0.16–2.39) |
| previous infection of SARS-CoV-2 | 0.17 | 0.873 | (-0.75–0.86) |
| fever | 2.01 | 0.079 | (-2.46–3.38) |
| vomiting | 0.80 | 0.452 | (-0.46–0.92) |
| diarrhea | 1.18 | 0.303 | (-0.95–2.36) |
| jaundice | -0.09 | 0.930 | (-1.86–1.72) |
| adenovirus | 0.25 | 0.808 | (-1.20–1.51) |
| liver transplantation | 2.24 | 0.045 | (0.02–1.47) |
| upper respiratory symptoms | 0.33 | 0.795 | (-29.45–31.04) |
| abdominal pain | -0.26 | 0.814 | (-1.50–1.28) |

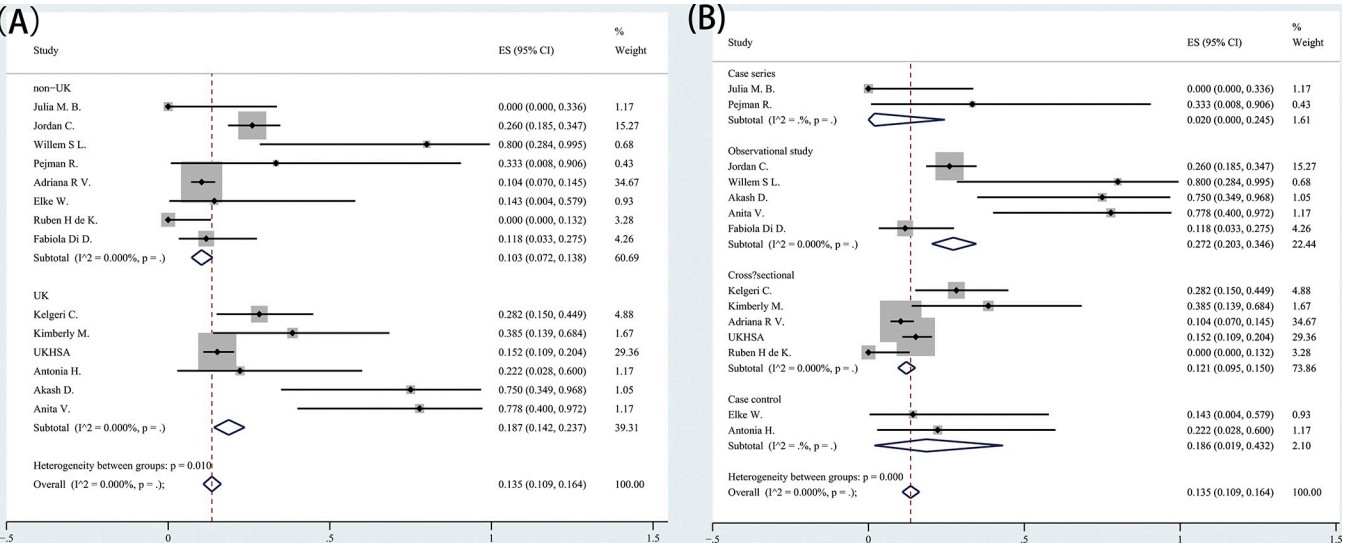

**Fig 5. Subgroup analysis.** (A) Subgroup analysis of area; (B) Subgroup analysis of study design.

the positive rate for SARS-CoV-2 in a study of Adriana et al. was 10.4%, while in a study of Deirdre et al [21].was reported as 18% among UK cases [27], which were slightly lower than our study. For the positive rate of adenovirus and co-infected with both pathogens, Adriana et al [21]. performed the incidence was 53.5% and 10.3% respectively, which were similar to the our study. A detailed investigation of cases in England in previous studies failed to identify common features of AS-Hep-UA in terms of exposure to drugs or toxins, pet contact, family structure, parental occupation, dietary or water sources, and so on [28].

nCoV has been a major public health threat since December 2019, of which the association with AS-Hep-UA is unknown to date, but it appeared that patients with COVID-19 were prone to abnormal liver function. Examining liver enzymes or biomarkers, e.g. ALT, AST, WBC, cytokines and GRP78, may contribute to differential diagnosis, but the specificity issues of these biomarkers should be carefully considered [29]. A study showed that 76.3% of patients with COVID-19 have abnormal liver test results and there was a 21.5% chance of developing liver injury during hospitalization. Patients with elevated liver enzymes classified as hepatocellular types during hospitalization had higher odds of progressing to severe disease [30]. Similarly, a published meta-analysis included nine researches on COVID-19 have shown that the incidence of liver injury was 27% (95% CI 18.2–35.8, I2 = 97%) [31]. For children with COVID-19,Up to 10% of children may be able to present with symptoms of hepatitis, based on previous reports [32]. However, gastrointestinal and hepatic manifestations were not exceptional in COVID-19, and diarrhea as well as mildly elevated liver enzymes were apparently the most common feature in the absent of preexisting COVID-19 [33].

The rarity and sporadic occurrence of these cases has obstructed systematic analysis to identify potential causes of AS-Hep-UA. However, this hepatitis outbreak was assumed to be associated with a deficiency of natural immunity to widespread childhood viruses secondary to "lockdown" and isolation during the COVID-19 pandemic [34]. Residues of SARS-CoV-2 have been detected in the gut in adults as long as 2 months after infection, and adults with "long-term COVID" exhibit an elevated level of interferon for several months after infection [32], base on which, some study believed that persistent SARS-CoV-2 infection in the intestine might result in the repeated release of viral proteins in the intestinal epithelium, leading to immune activation [27,28]. This repetitive immune activation may be mediated by a

superantigen motif in the SARS-CoV-2 spike-in proteins comparable to staphylococcal entero-toxin B, triggering broad and non-specific T-cell activation. It has been proposed that this superantigen-mediated immune cell activation could be the causative mechanism of the multi-systemic inflammatory syndrome(MIS)in children too [32]. In addition, a study found that SARS-CoV-2 virus may bind to angiotensin converting enzyme 2 (ACE2) which may be a potential target for viruses on bile duct cells, leading to bile duct cell dysfunction and eliciting a systemic inflammatory response leading to liver injury [35]. However, the UK Health Security Agency has stated in its latest report that 11.9% cases had a COVID-19 positive test, compared to 15.6% in the random sample–the difference was not statistically significant. This evidence, along with data published by researchers at the University of Glasgow and University College London, indicated that the increase in AS-Hep-UA was unlikely to be related to prior COVID-19 infection [36].

It has been shown that adeno-associated virus (AAV)is associated with the pathogenesis of AS-Hep-UA. AAV comes from a family of microviruses and the presence of another virus is required to facilitate its replication, usually adenovirus. And the peak of adenovirus infection in the community occurred just following an COVID-19 pandemic, which could be a strong point of support, providing powerful indirect evidence that combined infection with AAV2 and adenovirus may provoke the immune response leading to acute hepatitis [34]. However, it has also been shown that there was no evidence for the presence of viral inclusions in liver cells. Likewise, adenovirus has not been isolated by PCR with tissue from the liver, despite sustained positivity in the blood [34]. Moreover, it has also been put forward that the persistent immunosensitization of SARS-CoV-2 to the stinger protein superantigen in intestines, together with adenovirus intestinal infection, may lead to IFNγ release as well as IFNγ-mediated apoptosis of hepatocytes.

Although the cause of AS-Hep-UA remains elusive until now, thankfully, most patients have recovered with the conservative drug treatment, while only a small number of children require liver transplantation. Children are usually susceptible and high-risk groups for emerging infectious diseases, and the prevention and control of the transmission and epidemiology of emerging infectious diseases in children is a long-term task of basic and clinical medicine. In this sense, this retrospective systematic analysis of the relationship between acute severe hepatitis of unknown aetiology and adenovirus or SARS-CoV-2 in children can raise awareness of infectious diseases in the process of co-evolution of humans with pathogenic microorganisms.

## Strength and limitation

This meta-analysis summarized the symptom incidence and epidemiology of AS-Hep-UA, as well as systematically analyzed the relationship between AS-Hep-UA and SARS-CoV-2, adenovirus, which are beneficial to further understanding of AS-Hep-UA and provide some help for clinical work. However, our study has the following limitations: First, the included articles did not detail the specific strains isolated and it was not possible to pinpoint which strain of COVID-19 is highly correlated in children with AS-Hep-UA. Secondly, for countries with multiple reporting hospitals, severe cases may have been counted more than once because we did not have a unique identifier for cases, despite we excluded some duplicate populations by comparing detailed information. Thirdly, Some hospitals reported difficulties in retrieving case data, despite the use of electronic hospital databases. Fourth, some countries may have fewer reports of hepatitis due to a lack of awareness of AS-Hep-UA leading to regional bias in our reporting. Therefore, a review on how this type of data can be made available in the future in the light of epidemic and pandemic preparedness are pending completion.

## Conclusions

We summarized the incidence of symptoms in AS-Hep-UA and concluded from our study together with published articles that the positive rate of adenovirus was higher than SARS-CoV-2,and the relationship between AS-Hep-UA and COVID-19 is not significant because the positive rate of SARS-CoV-2 in AS-Hep-UA is relatively low, either current or previous infection. However, it cannot be excluded that the COVID-19 epidemic is an indirect causative agent of AS-Hep-UA, which requires systematic long-term follow-up in a larger cohort of AS-Hep-UA patients to uncover additional findings.

## Supporting information

**S1 Fig. Figures of egger test.**
(DOCX)

**S1 Table. Evaluation of articles by the case series study quality assessment tool.**
(DOCX)

**S2 Table. Evaluation of articles by the Newcastle-Ottawa Quality Assessment Scale.**
(DOCX)

**S1 File. Search strategy.**
(DOCX)

**S2 File. Checklist.**
(DOC)

**S3 File. Numbered table of all studies.**
(XLS)

## Author Contributions

**Conceptualization:** Jiayi Shan, Ran Li, Hua Xu.

**Data curation:** Yijun Xin, Xiaoling Zhang.

**Formal analysis:** Jiayi Shan, Ran Li.

**Investigation:** Jiayi Shan, Yijun Xin, Ran Li, Xiaoling Zhang.

**Methodology:** Jiayi Shan, Baoyi Huang, Yijun Xin, Xiaoling Zhang.

**Resources:** Baoyi Huang, Yijun Xin, Ran Li.

**Software:** Baoyi Huang.

**Writing – original draft:** Jiayi Shan, Ran Li.

**Writing – review & editing:** Jiayi Shan, Hua Xu.

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
