## [Decision Letter · Decision Letter 0]

22 Jul 2024

PONE-D-23-29634The clinical characteristics and SARS‐CoV‐2 infection in children of acute hepatitis with unknown aetiology: A meta‐analysis and systematic reviewPLOS ONE

Dear Dr. Xu,

Thank you for submitting your manuscript to PLOS ONE. After careful consideration, we feel that it has merit but does not fully meet PLOS ONE’s publication criteria as it currently stands. Therefore, we invite you to submit a revised version of the manuscript that addresses the points raised during the review process.

We look forward to receiving your revised manuscript.

Kind regards,

Benjamin M. Liu, MBBS, PhD, D(ABMM), MB(ASCP)

Academic Editor

PLOS ONE

Journal Requirements:

2. We note that this manuscript is a systematic review or meta-analysis; our author guidelines therefore require that you use PRISMA guidance to help improve reporting quality of this type of study. Please upload copies of the completed PRISMA checklist as Supporting Information with a file name “PRISMA checklist”.

https://onlinelibrary.wiley.com/doi/10.1002/jmv.26314

https://pubmed.ncbi.nlm.nih.gov/35551703/

In your revision ensure you cite all your sources (including your own works), and quote or rephrase any duplicated text outside the methods section. Further consideration is dependent on these concerns being addressed.

4. We note that your Data Availability Statement is currently as follows: Data availability is not applicable to this article as no new data were created or analyzed in this study.

**Additional Editor Comments:**

Editor's comments:

Please add the following sentence and new references to Discussion: "Examining liver enzymes or biomarkers, e.g. ALT, AST, cytokines and GRP78, may contribute to differential diagnosis, but the specificity issues of these biomarkers should be carefully considered". More references are suggested to cite, with the following ref as examples (citing is optional).

Liu B, Yang JX, Yan L, Zhuang H, Li T. Novel HBV recombinants between genotypes B and C in 3'-terminal reverse transcriptase (RT) sequences are associated with enhanced viral DNA load, higher RT point mutation rates and place of birth among Chinese patients. Infect Genet Evol. 2018 Jan;57:26-35. doi: 10.1016/j.meegid.2017.10.023. Epub 2017 Oct 27. PMID: 29111272.

Peng Y, Liu B, Hou J, Sun J, Hao R, Xiang K, Yan L, Zhang J, Zhuang H, Li T. Naturally occurring deletions/insertions in HBV core promoter tend to decrease in hepatitis B e antigen-positive chronic hepatitis B patients during antiviral therapy. Antivir Ther. 2015;20(6):623-32. doi: 10.3851/IMP2955. Epub 2015 Apr 2. PMID: 25838313.

Liu BM, Beck EM, Fisher MA. The Brief Case: Ventilator-Associated Corynebacterium accolens Pneumonia in a Patient with Respiratory Failure Due to COVID-19. J Clin Microbiol. 2021 Aug 18;59(9):e0013721. doi: 10.1128/JCM.00137-21. Epub 2021 Aug 18. PMID: 34406882; PMCID: PMC8372998.

Wei D, Li NL, Zeng Y, Liu B, Kumthip K, Wang TT, Huo D, Ingels JF, Lu L, Shang J, Li K. The Molecular Chaperone GRP78 Contributes to Toll-like Receptor 3-mediated Innate Immune Response to Hepatitis C Virus in Hepatocytes. J Biol Chem. 2016 Jun 3;291(23):12294-309. doi: 10.1074/jbc.M115.711598. Epub 2016 Apr 20. PMID: 27129228; PMCID: PMC4933277.

Reviewers' comments:

Reviewer's Responses to Questions

**Comments to the Author**

1. Is the manuscript technically sound, and do the data support the conclusions?

Reviewer #1: Yes

2. Has the statistical analysis been performed appropriately and rigorously? 

Reviewer #1: Yes

3. Have the authors made all data underlying the findings in their manuscript fully available?

Reviewer #1: Yes

4. Is the manuscript presented in an intelligible fashion and written in standard English?

Reviewer #1: Yes

5. Review Comments to the Author

Reviewer #1: Acute severe hepatitis of unknown aetiology in children presents with a high number of similar cases in a short period of time, and some of them are accompanied by fever, suggesting the possibility of an infectious etiology.

Although there is a lack of a clear epidemiologic association of the existing cases, whether adenoviruses or other pathogens undergo histophilic and pathogenic changes or are novel deserves attention and focus.

In this sense, the retrospective systematic analysis of the relationship between acute severe hepatitis of unknown aetiology and adenovirus or SARS-CoV-2 in children in this paper can raise awareness of infectious diseases in the process of co-evolution of humans with pathogenic microorganisms.

Children are usually susceptible and high-risk groups for emerging infectious diseases, and the prevention and control of the transmission and epidemiology of emerging infectious diseases in children is a long-term task of basic and clinical medicine.

6. PLOS authors have the option to publish the peer review history of their article (what does this mean?). If published, this will include your full peer review and any attached files.

Reviewer #1: No

---

## [Author Response · Author response to Decision Letter 0]

29 Aug 2024

1.Please ensure that your manuscript meets PLOS ONE's style requirements, including those for file naming. 

#I have changed the manuscript according to PLOS ONE's style.

2.We note that this manuscript is a systematic review or meta-analysis; our author guidelines therefore require that you use PRISMA guidance to help improve reporting quality of this type of study. Please upload copies of the completed PRISMA checklist as Supporting Information with a file name “PRISMA checklist”.

#I have upload copies of the completed PRISMA checklist as Supporting Information with a file name “PRISMA checklist”.

https://onlinelibrary.wiley.com/doi/10.1002/jmv.26314

https://pubmed.ncbi.nlm.nih.gov/35551703/

#I have cited the above 2 articles, see Ref.26 and Ref.32

4. We note that your Data Availability Statement is currently as follows: Data availability is not applicable to this article as no new data were created or analyzed in this study.

#I have upload minimal data set as Supporting Information with a file name “minimal data set”.

5.Please include captions for your Supporting Information files at the end of your manuscript, and update any in-text citations to match accordingly.

#I have upload captions of my Supporting Information at the end of my manuscript.

6.Please review your reference list to ensure that it is complete and correct. 

#Reference list has been checked.

---

## [Editor Report · Decision Letter 1]

2 Sep 2024

PONE-D-23-29634R1The clinical characteristics and SARS‐CoV‐2 infection in children of acute hepatitis with unknown aetiology: A meta‐analysis and systematic review

PLOS ONE

Dear Dr. Xu,

Thank you for submitting your manuscript to PLOS ONE. After careful consideration, we feel that it has merit but does not fully meet PLOS ONE’s publication criteria as it currently stands. Therefore, we invite you to submit a revised version of the manuscript that addresses the points raised during the review process.

The authors failed to address the Editor's and the Reviewer #1's comments listed in the prior decision letter. But the Editor would like to give the authors the last opportunity to address these comments. 

Editor's comments:

Please add the following sentence and new references to Discussion: "Examining liver enzymes or biomarkers, e.g. ALT, AST, WBC, cytokines and GRP78, may contribute to differential diagnosis, but the specificity issues of these biomarkers should be carefully considered". More references are suggested to cite, with the following ref as examples (citing is optional).

Liu B, Yang JX, Yan L, Zhuang H, Li T. Novel HBV recombinants between genotypes B and C in 3'-terminal reverse transcriptase (RT) sequences are associated with enhanced viral DNA load, higher RT point mutation rates and place of birth among Chinese patients. Infect Genet Evol. 2018 Jan;57:26-35. doi: 10.1016/j.meegid.2017.10.023. Epub 2017 Oct 27. PMID: 29111272.

Peng Y, Liu B, Hou J, Sun J, Hao R, Xiang K, Yan L, Zhang J, Zhuang H, Li T. Naturally occurring deletions/insertions in HBV core promoter tend to decrease in hepatitis B e antigen-positive chronic hepatitis B patients during antiviral therapy. Antivir Ther. 2015;20(6):623-32. doi: 10.3851/IMP2955. Epub 2015 Apr 2. PMID: 25838313.

Liu BM, Beck EM, Fisher MA. The Brief Case: Ventilator-Associated Corynebacterium accolens Pneumonia in a Patient with Respiratory Failure Due to COVID-19. J Clin Microbiol. 2021 Aug 18;59(9):e0013721. doi: 10.1128/JCM.00137-21. Epub 2021 Aug 18. PMID: 34406882; PMCID: PMC8372998.

Wei D, Li NL, Zeng Y, Liu B, Kumthip K, Wang TT, Huo D, Ingels JF, Lu L, Shang J, Li K. The Molecular Chaperone GRP78 Contributes to Toll-like Receptor 3-mediated Innate Immune Response to Hepatitis C Virus in Hepatocytes. J Biol Chem. 2016 Jun 3;291(23):12294-309. doi: 10.1074/jbc.M115.711598. Epub 2016 Apr 20. PMID: 27129228; PMCID: PMC4933277.

Reviewer #1: Acute severe hepatitis of unknown aetiology in children presents with a high number of similar cases in a short period of time, and some of them are accompanied by fever, suggesting the possibility of an infectious etiology.

Although there is a lack of a clear epidemiologic association of the existing cases, whether adenoviruses or other pathogens undergo histophilic and pathogenic changes or are novel deserves attention and focus.

In this sense, the retrospective systematic analysis of the relationship between acute severe hepatitis of unknown aetiology and adenovirus or SARS-CoV-2 in children in this paper can raise awareness of infectious diseases in the process of co-evolution of humans with pathogenic microorganisms.

Children are usually susceptible and high-risk groups for emerging infectious diseases, and the prevention and control of the transmission and epidemiology of emerging infectious diseases in children is a long-term task of basic and clinical medicine.

We look forward to receiving your revised manuscript.

Kind regards,

Benjamin M. Liu, MBBS, PhD, D(ABMM), MB(ASCP)

Academic Editor

PLOS ONE

---

## [Author Response · Author response to Decision Letter 1]

12 Sep 2024

1.Editor's comments:Please add the following sentence and new references to Discussion: "Examining liver enzymes or biomarkers, e.g. ALT, AST, WBC, cytokines and GRP78, may contribute to differential diagnosis, but the specificity issues of these biomarkers should be carefully considered". More references are suggested to cite, with the following ref as examples (citing is optional).

#I have added the above sentence and new references to Discussion.see text maked in red.

2.Reviewer #1: Acute severe hepatitis of unknown aetiology in children presents with a high number of similar cases in a short period of time, and some of them are accompanied by fever, suggesting the possibility of an infectious etiology.

#I have modified the discussion section based on Review1's suggestion.see text maked in red.

---

## [Editor Report · Decision Letter 2]

16 Sep 2024

The clinical characteristics and SARS‐CoV‐2 infection in children of acute hepatitis with unknown aetiology: A meta‐analysis and systematic review

PONE-D-23-29634R2

Dear Dr. Xu,

We’re pleased to inform you that your manuscript has been judged scientifically suitable for publication and will be formally accepted for publication once it meets all outstanding technical requirements.

Kind regards,

Benjamin M. Liu, MBBS, PhD, D(ABMM), MB(ASCP)

Academic Editor

PLOS ONE
---

## [Editor Report · Acceptance letter]

27 Sep 2024

PONE-D-23-29634R2 

PLOS ONE

Dear Dr. Xu, 

I'm pleased to inform you that your manuscript has been deemed suitable for publication in PLOS ONE. Congratulations! Your manuscript is now being handed over to our production team.

Kind regards, 

on behalf of

Dr. Benjamin M. Liu 

Academic Editor

PLOS ONE